# FIREWORKS: a bottom-up approach to integrative coessentiality network analysis

David R Amici[1,2,3,4], Jasen M Jackson[1,2,3], Mihai I Truica[3,4,5] , Roger S Smith[1,2,3,4] , Sarki A Abdulkadir[3,5], Marc L Mendillo[1,2,3]

Genetic coessentiality analysis, a computational approach which identifies genes sharing a common effect on cell fitness across large-scale screening datasets, has emerged as a powerful tool to identify functional relationships between human genes. However, widespread implementation of coessentiality to study individual genes and pathways is limited by systematic biases in existing coessentiality approaches and accessibility barriers for investigators without computational expertise. We created FIREWORKS, a method and interactive tool for the construction and statistical analysis of coessentiality networks centered around gene(s) provided by the user. FIREWORKS incorporates a novel bias reduction approach to reduce false discoveries, enables restriction of coessentiality analyses to custom subsets of cell lines, and integrates multiomic and drug–gene interaction datasets to investigate and target contextual gene essentiality. We demonstrate the broad utility of FIREWORKS through case vignettes investigating gene function and specialization, indirect therapeutic targeting of "undruggable" proteins, and context-specific rewiring of genetic networks.

## Introduction

Now 20 yr removed from the first draft of the human genome, our understanding of how genes function together to form cellular and organismal networks is still growing rapidly. Genetic coessentiality analysis, a guilt-by-association computational approach rooted in principles from foundational studies in yeast (Boone et al, 2007), has recently emerged as a powerful tool to infer the function of human genes as well as the relationships between them. Coessentiality analysis is based on the observation that the importance of a given gene to cellular growth (or any other phenotype) depends on cellular context (Rancati et al, 2018). That is, factors such as genetic background, tissue of origin, and cell culture conditions can

all impact the "essentiality" of a given gene (defined here as a continuous variable reflecting a gene's importance for cellular growth). Thus, by quantifying the essentiality of each protein-coding gene across hundreds of biological contexts—such as established cancer cell lines derived from unique tumors—genes with highly similar fitness variations across contexts may be identified and considered putatively co-functional. The observation that strong genetic fitness correlations are predictive of participation in the same biological process has already spurred discoveries of novel gene functions from publicly available, genome-scale fitness screening datasets (McDonald et al, 2017; Wang et al, 2017; Boyle et al, 2018; Pan et al, 2018; Rauscher et al, 2018; Kim et al, 2019; Wainberg et al, 2019 *Preprint*; Bayraktar et al, 2020).

Despite the great potential of coessentiaity analysis to improve our understanding of the human genome, several factors limit its widespread adoption. Existing coessentiality resources take a "top-down" approach, whereby the strongest of all possible gene–gene correlations are clustered at genome-scale. Although powerful for detecting high-confidence interactions, these top-down approaches predominantly yield clusters of genes which function in obligate cooperativity (e.g., those encoding members of the same protein complex) or which represent technical artifacts (e.g., genes located on the same chromosome). In turn, much of the genome does not belong to informative clusters in these approaches. Even with inevitable algorithmic improvements in the identification and clustering of coessential fitness profiles, many genes with multifaceted functions may not be expected to ever segregate into one module. Indeed, many crucial genes which have a dynamic network of effectors and affect multiple biological processes—such as transcription factors, E3 ligases, and kinases—are unlikely to ever attain the strong and reciprocal coessentiality phenotype needed to form a distinct cluster in top-down coessentiality networks. As such, "bottom-up" coessentiality networks centered upon these complicated, but critically important, genes may better provide insight into their functional relationships. In addition to generalized limitations of top-down network analysis, there is a broader need for methods to seamlessly integrate orthogonal

[1]Department of Biochemistry and Molecular Genetics, Northwestern University, Chicago, IL, USA   [2]Simpson Querrey Center for Epigenetics, Northwestern University Feinberg School of Medicine, Chicago, IL, USA   [3]Robert H Lurie Comprehensive Cancer Center, Northwestern University Feinberg School of Medicine, Chicago, IL, USA   [4]Medical Scientist Training Program, Northwestern University Feinberg School of Medicine, Chicago, IL, USA   [5]Department of Urology, Northwestern University Feinberg School of Medicine, Chicago, IL, USA

Correspondence: mendillo@northwestern.edu

datasets with genetic coessentiality data. For example, by integrating multiomic characterization data for the cell lines used in coessentiality analyses, one may gain insight into the contextual factors which drive dependence on a given gene or functional module. Finally, and perhaps most importantly, intuitive tools are required to overcome the accessibility barrier which currently limits custom coessentiality network analyses to those with computational and domain expertise.

Here, we introduce FIREWORKS (Fitness Interaction Rank-Extrapolated netWORKs; fireworks.mendillolab.org), an interactive web tool for customizable, bottom-up coessentiality network analysis constructed around the user's gene(s) of interest (Fig 1). FIREWORKS addresses the principal remaining source of systematic bias in CRISPR-based coessentiality analyses, implements novel features such as context-specific coessentiality networks and orthogonal data integration, and requires no coding or subject matter expertise.

# Results

### A sliding window preprocessing approach to reduce correlation bias from genes in close physical proximity

Previous CRISPR-based coessentiality analyses have reported an overrepresentation of highly coessential genes which share no known biological function but are located in the same chromosomal neighborhood (Boyle et al, 2018; Pan et al, 2018; Kim et al, 2019; Bayraktar et al, 2020). This bias is thought to reflect a phenotypic artifact of Cas9-mediated DNA cleavage in copy number–variable regions which is not entirely addressed by the best normalization and de-noising approaches currently available (Dempster et al, 2019 *Preprint*). We sought to quantify the extent to which locus bias affects CRISPR-based coessentiality estimates for individual genes across the genome and develop a preprocessing approach that mitigates this source of error. To determine the expected rate of syntenic (i.e., same chromosome) coessentiality, we considered two null distributions: the first attributable to chance based on the number of genes on each chromosome (i.e., larger chromosomes have higher expected syntenic coessentiality rates; "Random") and the second from coessentiality analysis performed with data from 712 shRNA genetic screens, which do not produce a DNA damage phenotype when targeting copy number–variable genes ("RNAi") (McFarland et al, 2018). The latter would be expected to identify any overrepresentation of coessentiality signal from bona fide co-functional neighbor genes. Indeed, estimates of syntenic coessentiality in RNAi data largely matched those expected from random chance, with few positive outlier exceptions often reflecting duplicated genes which retain putatively similar functions and thus loss-of-function fitness profiles (Figs 2A and S1A).

We next identified the proportion of each gene's top 100 ranked fitness correlations which are located on the same chromosome using CRISPR-Cas9 gene essentiality estimates derived from the

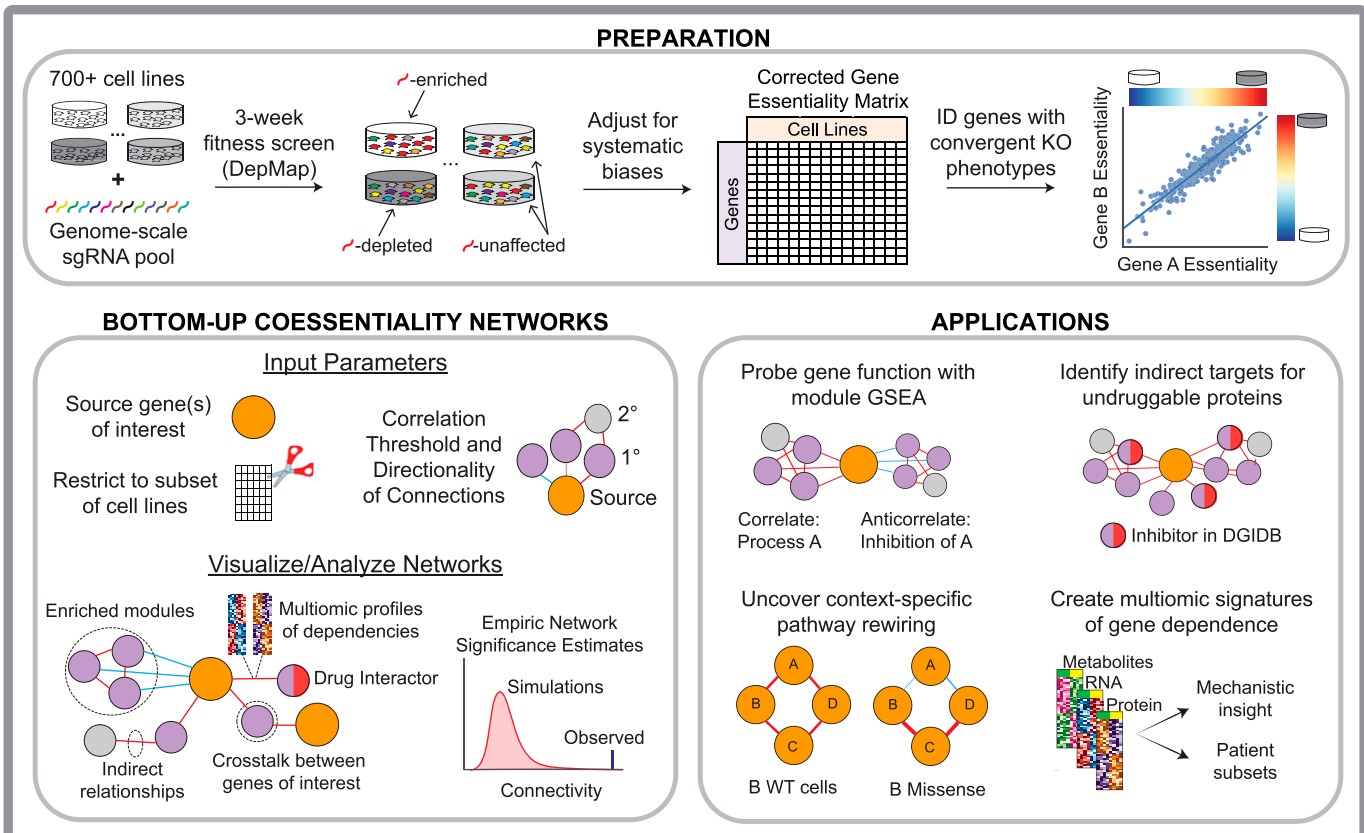

**Figure 1.  Schematic representation of bottom-up, integrative coessentiality network mapping with FIREWORKS.**

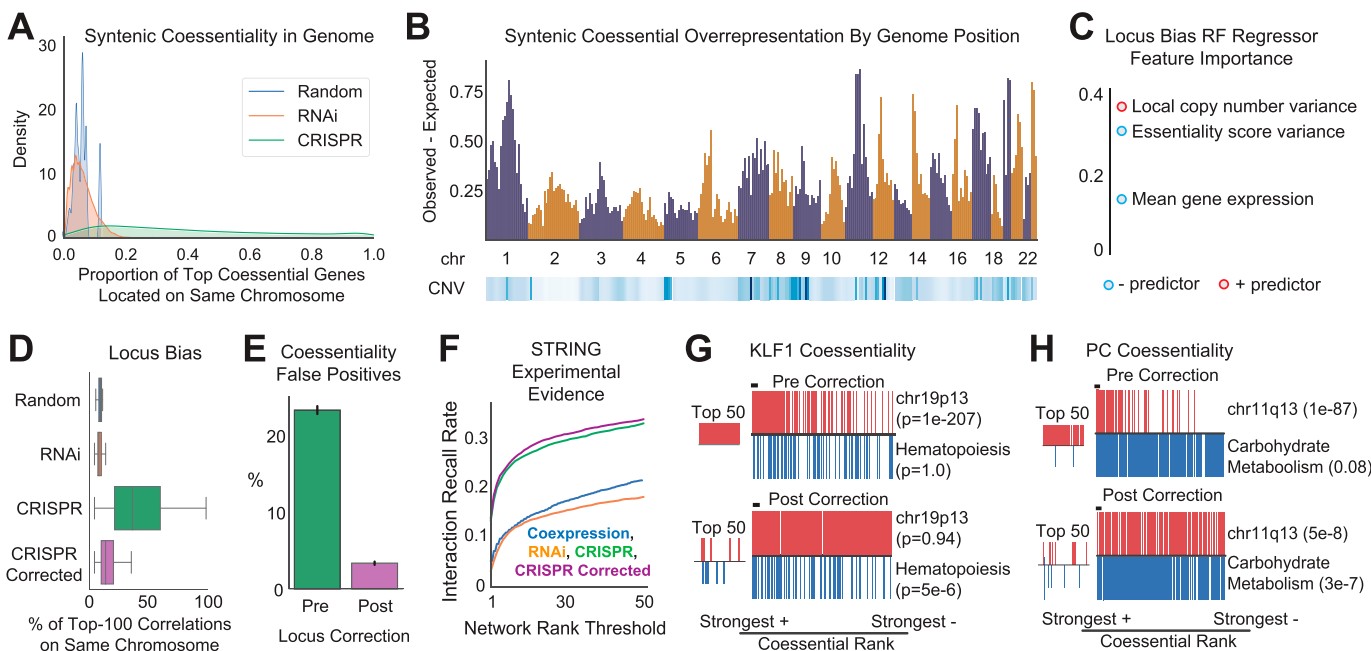

**Figure 2.  Correction of genomic locus bias reduces false positives and increases predictive power in CRISPR coessentiality analysis.**
**(A)** Genome-scale distribution of the fraction of each gene's top 100 ranked fitness correlations which are co-localized in the same chromosomal band. Random indicates frequency based on chromosome gene content, whereas RNAi indicates coessentiality computed using shRNA screen data. CRISPR data and RNAi data stem from 739 and 712 cell lines, respectively. **(B)** Median locus bias (syntenic coessentiality rate observed minus maximum expected from RNAi coessentiality or random chance) and copy number variability (CNV; blue is higher variability) for chromosomal band neighborhoods across the genome. **(C)** Gini importance, a measure of the power of a feature to reduce model uncertainty, of gene-level features in a Random Forest regression model trained to predict locus bias. **(D)** The neighbor subtraction preprocessing approach for locus correction (see the Materials and Methods section and Fig S2) reduces the burden of locus-biased false positives in CRISPR coessentiality analysis. **(E)** Presumed false positives (syntenic correlations beyond threefold expected by either chance or RNAi coessentiality) comprise 23% and 3% of the average gene's top 50 ranked correlations before and after correction, respectively. **(F)** Locus-corrected CRISPR coessentiality data identifies more true positive experimental interactions than non-corrected CRISPR coessentiality, RNAi coessentiality, and transcript co-expression datasets. **(G, H)** The coessentiality profile of highly locus-biased genes before and after locus correction reveals increased prioritization of known relationships and a reduction in locus-associated false positives. *P*-value from hypergeometric test.

Broad Institute's Dependency Map screening project of 739 cancer cell lines (Meyers et al, 2017; Tsherniak et al, 2017) (Fig 2A). Importantly, these CRISPR-based essentiality estimates have already undergone several bias reduction steps, including application of the CERES algorithm (which adjusts the gene effect estimate based upon local copy number) and principal components analysis (PCA)-based denoising similar to that described by Boyle et al (Meyers et al, 2017; Boyle et al, 2018; Dempster et al, 2019 *Preprint*). Despite these preprocessing steps, we found evidence of substantial locus bias in CRISPR coessentiality estimates, with the median gene in the CRISPR dataset having ~6-fold more syntenic correlations than the largest expected value from random chance or RNAi coessentiality (Fig 2A). Remarkably, more than 1,000 genes had every one of their top-ranked correlations located on the same chromosome, and more specifically, within the same chromosomal band region. The bands containing the highest numbers of locus-biased genes often represented genomic regions commonly amplified in cancer (Fig S1B). However, by quantifying the median overrepresentation of neighbor genes and copy number variance of each chromosomal band neighborhood, we determined that locus bias spans the genome and is not entirely explained by local copy number variability (Fig 2B). To better understand the driving factors of locus bias in CRISPR coessentiality, we trained a Random Forest regression

model to predict locus bias from gene-level features: gene expression in cancer cell lines (mean and minimum), essentiality score in Achilles CRISPR screens (variance, mean, and maximum), copy number variance of the gene and its chromosomal band, gene length, and duplicate gene status. This model yielded a decision tree which primarily used three features to predict locus bias with ~17% mean absolute error: local copy number variance, essentiality score variance, and mean gene expression (Figs 2C and S1C–E). In sum, our model predicts that genes with weaker or less variable fitness effects in CRISPR knockout screens are more likely to have higher levels of locus bias, particularly if they are located within genomic regions subject to copy number variation.

Next, we tested the impact of several preprocessing approaches designed to reduce the overrepresentation of neighbor genes in CRISPR coessentiality analysis. These approaches included removal of principal components explaining essentiality score variance in highly locus-biased genes, normalization of gene-level correlations by considering neighbor genes' fitness scores, and penalization of correlations between genes on the same chromosome (see the Materials and Methods section and Fig S2A–C). We highlight the best-performing approach: "neighbor subtraction," a sliding window correction similar to that described previously (Wang et al, 2017) where, for each pre-correction gene not located within a

duplicate gene cluster, half the median essentiality score of 40 neighbor genes is subtracted from the pre-correction gene's initial essentiality estimate (Fig S2C). We note that these adjustment parameters (e.g., number of neighbors) were determined through unbiased benchmarking (Materials and Methods section). The sliding window correction approach serves to effectively smooth out fitness effects of targeting a given chromosomal locus while preserving the relative fluctuations in essentiality directly attributable to the target gene. Critically, neighbor subtraction preprocessing substantially reduced the abundance of syntenic correlations in CRISPR coessentiality analysis (Figs 2D and S2A). Stated differently, false positive coessential relationships–defined for this purpose as syntenic correlations in threefold excess of those expected from chance or RNAi coessentiality—made up 23% of the average gene's top-ranked correlations before correction, but only 3% after correction (Fig 2E).

To ensure that removal of systematic locus bias through neighbor subtraction did not reduce the ability of CRISPR coessentiality to identify true positive interactions, we benchmarked our corrected coessentiality dataset against curated interaction databases (i.e., CORUM protein complex members, STRING high-confidence experimental interactions, and gene set enrichment analysis [GSEA] pathway gene sets). We find that locus correction did not reduce the ability of CRISPR coessentiality to robustly detect true biological interactions, instead conferring a modest improvement in predictive power (Figs 2F and S2B). Importantly, locus correction yielded meaningful improvements even for the most biased genes, such as the 1,019 genes which had 100% syntenic coessentiality amongst their top coessential relationships before correction. For example, the top pre-correction correlations for PC (pyruvate carboxylase; the enzyme which converts pyruvate to oxaloacetate) and KLF1 (a hematopoietic lineage transcription factor) comprised only genes located in close physical proximity on the same chromosome. However, after correction, the top correlations for these genes were enriched for carbohydrate metabolism and hematopoietic differentiation, respectively (both overlap $P$-values < $1 \times 10^{-5}$) (Fig 2G and H). Altogether, these data indicate that locus bias correction robustly improves the signal-to-noise ratio in CRISPR-based coessentiality analyses.

## Construction of bottom-up genetic networks reveals insight into gene function and specialization

To expand the reach of coessentiality analysis to the whole genome, we sought to use our locus-bias-adjusted coessentiality matrix to create a compendium of "bottom-up" coessentiality networks centered upon each protein-coding gene in the genome. Briefly, for each gene, the top 30 ranked correlations and anti-correlations were incorporated as "primary nodes" in an undirected, unweighted network. Then, the top five correlations for each primary node were determined, with any of these genes connected to multiple primary nodes being incorporated in the network as "secondary nodes" which serve to functionally cluster the primary nodes into functional modules. In general, we found the rank thresholds of 30 (primary) and 5 (secondary, positive only) to best reveal biological signal while maintaining visibility of individual genes. However, rank parameters are entirely customizable in the

FIREWORKS web portal. As a control, we created "noise" fitness profiles for 10,000 simulated genes by randomly sampling essentiality scores observed in the Project Achilles screening dataset and then subjected these simulated genes to the same network construction process. At an empiric false discovery rate cutoff of 0.5%, every protein-coding gene in the genome had significantly stronger associations with its primary nodes than could be attributable to chance, indicating that the bottom-up coessentiality network is broadly applicable and not limited to genes with strong or highly variable fitness effects (Fig 3A).

To investigate the organization and composition of bottom-up networks across the genome, we used Louvain's algorithm to group each gene's network into modules (Blondel et al, 2008). Briefly, this approach assigns genes to modules in a manner which maximizes the density of connections inside modules relative to connections outside modules. We found great heterogeneity in the degree to which networks segregated into highly interconnected modules (i.e., network modularity). Networks with low-modularity scores typically originated from specialized components of large molecular assemblies where all network components are thoroughly interwoven in one large community. Examples of low-modularity networks include NDUFAF8, MRPL11, and PEX26, which are components of mitochondrial complex I, the mitochondrial ribosome, and the peroxisome (Fig 3B and C)—obligate members of molecular machines which form large clusters in top-down network analyses (Boyle et al, 2018; Pan et al, 2018; Kim et al, 2019; Wainberg et al, 2019 Preprint). On the other hand, high-modularity networks often originated from genes with multifaceted roles in pathways dynamically regulated by multiple signals, such as RHEB, PPP1R15B, and GAPDH (Fig 3B). The dense modules within these bottom-up networks provide clear insight into the biology of the source node factors. For example, RHEB is negatively connected to a module containing the TSC1-TSC2 complex and positively connected to a module representing the mTORC1 complex, consistent with RHEB's known function to activate mTORC1 in a manner inhibited by TSC1/2 (Fig 3D). As another example, PPP1R15B encodes a phosphatase which functions to terminate the integrated stress response (ISR), a generalized remodeling of translation activated by mechanistically diverse stressors. The PPP1R15B network contains modules reflecting stressors known to feed into the ISR (nutrient, metabolic, and proteotoxic stress), proteins which directly activate the ISR (EIF2AK4 and ATF4), and effectors downstream of ISR termination (EIF2B4) (Fig S3). Altogether, these data indicate that bottom-up coessentiality network analysis is a viable and broadly applicable approach to gain insight into the biology of individual genes.

## Nomination of surrogate therapeutic targets for undruggable proteins

Many proteins implicated in human disease have structures which are challenging to target pharmacologically. For example, oncogenic drivers such as MYC and KRAS have remained largely impervious to targeted therapies despite decades of intensive, interdisciplinary research (Dang et al, 2017). We hypothesized that the coessentiality network of these challenging targets—which, by definition, contain genes particularly essential to cells dependent on that challenging target—would include genes which are

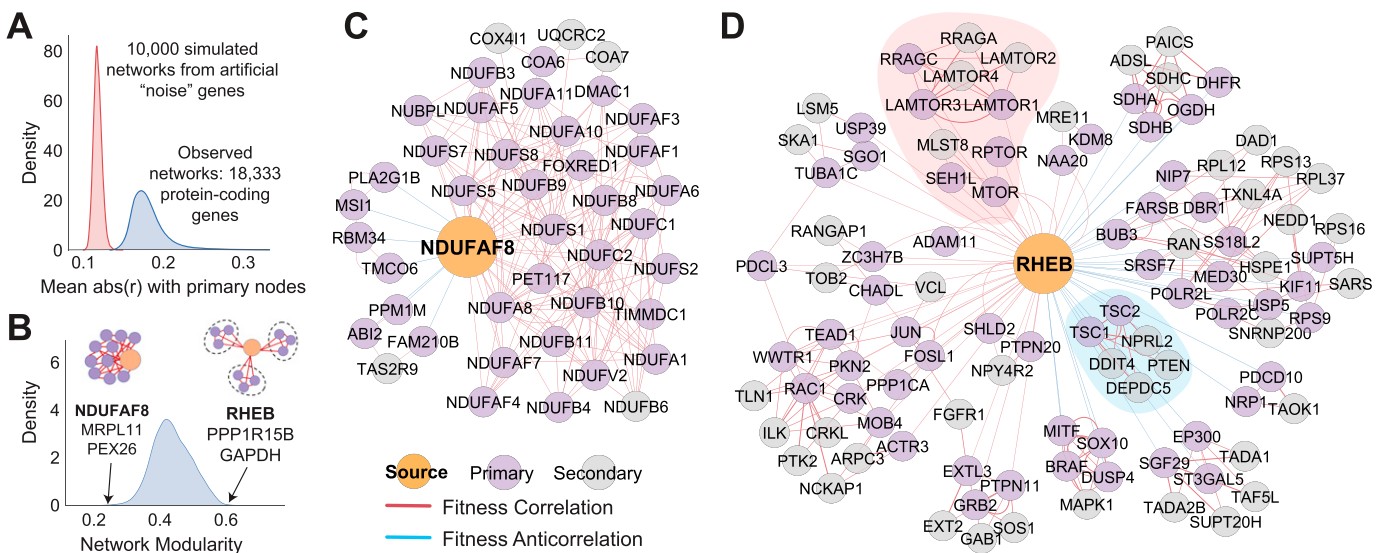

**Figure 3. Construction of a bottom-up coessentiality network for every gene in the genome.**
**(A)** A standard bottom-up coessentiality network, as described in text, was created for every gene in the genome as well as 10,000 simulated fitness profiles created from random sampling of gene essentiality data. The average absolute magnitude of the Pearson correlation for the primary connections observed from actual genes was all stronger than at least 99.5% of simulated networks. **(B)** Modularity of each gene's bottom-up coessentiality network after application of Louvain's algorithm for community detection. Examples of low-modularity and high-modularity networks are highlighted. **(C)** NDUFAF8, a component of complex I in the electron transport chain, is an example of a low-modularity network dominated by genes related to oxidative phosphorylation. **(D)** RHEB, a small GTPase involved in mTORC1 regulation, is an example of a high modularity network containing many clusters of densely interconnected genes. The red module represents genes involved in mTORC1 activation downstream of RHEB and the blue module represents the TSC1-TSC2 complex which negatively regulates RHEB to inactivate mTORC1 signaling. Note that double looping (two connections between a given gene pair) indicates that the correlation relationship is among the top-ranked for both genes at the specified rank thresholds (here, 30 for primary nodes and 5 for secondary nodes).

exploitable by available drugs and may thus serve as indirect targets. To facilitate rapid screening of coessentiality networks for genes containing known drug interactions, we integrated data from a drug–gene interaction database (Cotto et al, 2018) into our bottom-up coessentiality networks. Even at stringent rank thresholds, most networks across the genome contained genes targeted by existing drugs, many of which have reported mechanisms of action (Fig 4A). From this framework, we queried the networks of an array of attractive drug targets across fields such as cancer biology, aging, and neurodegeneration, finding that many were highly coessential with a gene targeted by existing drugs (Fig 4B and Table S1).

Supporting the validity of the coessentiality approach to nominate indirect therapeutic targets, several drugs identified by our integrative coessentiality analysis of attractive targets have been explicitly designed for this purpose or have demonstrated success in drug repurposing. For example, inhibitors of EGLN1, a protein which regulates the stability of Hypoxia Inducible Factor 1α (HIF1A) and is HIF1A's second-ranked anticorrelation, are being explored to activate a hypoxia response in the treatment of renal anemia (Maxwell & Eckardt 2016). As another example, RAF1/c-Raf, the direct downstream effector of KRAS and KRAS' top-ranked correlation, is the target of many inhibitors designed to hamper Ras-driven tumorigenesis (Burotto et al, 2014). Finally, inhibition of CDK7, a transcriptional cyclin-dependent kinase which functions downstream of MYC and is MYC's 10th-ranked correlation, causes marked regression of aggressive, MYC-driven neuroblastomas in mice (Chipumuro et al, 2014).

Critically, our approach also identifies several gene-drug associations which have not been previously explored and may thus represent novel therapeutic strategies. For example, to an even greater extent than CDK7, MYC-dependent cells are co-dependent on WNK1 (coessential rank 2; Fig 4C), a kinase inhibited by the small molecule PP121 (Yagi et al, 2009). Further supporting the possibility of targeting MYC-dependent cancer cells through WNK1 inhibition, we find evidence in an orthogonal drug repurposing screen (Corsello et al, 2020) that cells with high expression of MYC or dependence on MYC have increased sensitivity to PP121 treatment (Fig 4D). Finally, we experimentally validated the interaction between MYC and PP121, finding that MYC deletion decreases cellular sensitivity to PP121 in fibroblasts (Fig 4E). Altogether, these data demonstrate the power of integrative coessentiality analysis to uncover alternative therapeutic approaches for classically challenging drug targets.

### Context-specific rewiring of genetic networks uncovered by differential coessentiality

In general, coessentiality analyses encompassing all available cell lines provides the greatest representation of biological contexts, and thus the greatest power to uncover functional relationships between genes. However, genetic networks are dynamically regulated depending on cellular context (Bandyopadhyay et al, 2010). In turn, coessentiality signal for certain context-specific genetic interactions may be obscured by pooling fitness profiles from all cell lines. As an example, oncogenic driver mutations are known to rewire cellular signaling. For instance, downstream effectors in the

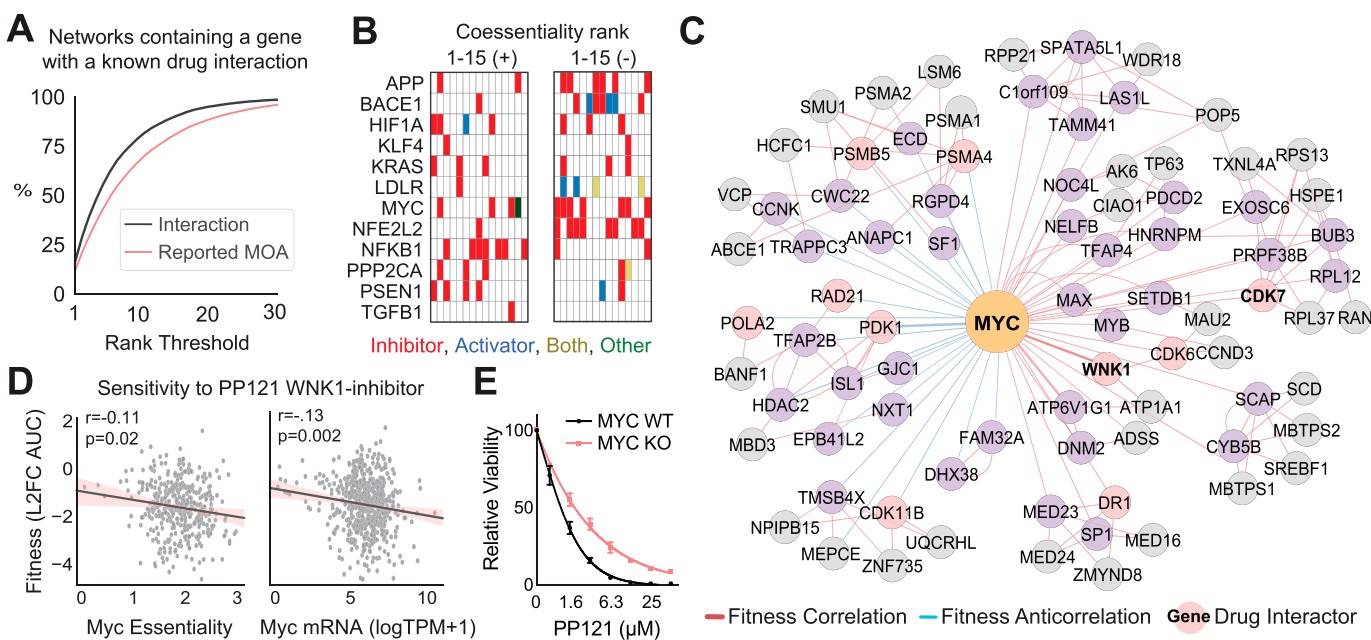

**Figure 4. Integration of drug–gene interaction data to identify surrogate therapeutic targets for challenging proteins.**
**(A)** Proportion of bottom-up coessentiality networks in the genome which contain at least one protein with a known gene-drug interaction in the Drug-Gene Interaction Database at the specified rank threshold. For (A), only positive primary nodes are considered. Reported mechanism of action refers to drug–gene interactions characterized with mechanisms such as "inhibitor" or "activator." **(B)** Presence of drug–gene interactions with reported mechanism of action for the top 15 ranked correlations and anticorrelations for a panel of attractive therapeutic target proteins. **(C)** An example bottom-up network for a challenging therapeutic target, MYC, which has a coessential knockout phenotype with several genes targeted by existing drugs (red nodes). **(D)** Cancer cell dependence on and expression of MYC are associated with greater sensitivity to the WNK inhibitor PP121. P-value from Pearson correlation. **(E)** Viability of MYC KO (HO15.19) or parental MYC WT (TGR-1) rat fibroblasts treated with PP121 at the indicated concentrations. Three biological replicates per dose.

MAPK pathway (e.g., Ras, Raf, and MEK; Fig 5A) are commonly subject to activating mutations in tumors in a manner which renders upstream epidermal growth factor receptor (EGFR) activation inconsequential for downstream MEK-ERK signaling (Burotto et al, 2014). Of the 739 cell lines used in our pan-cancer analyses, 96 contain a missense (presumed activating) mutation in BRAF. We thus computed a genome-scale correlation matrix for both BRAF-WT and BRAF-mutant cell lines. To identify coessential relationships specific to or enriched in the BRAF-mutant context, we generated a differential correlation matrix by subtracting the BRAF-WT matrix from the BRAF-mutant matrix.

Intriguingly, relationships between genes involved in the MAPK pathway were among the most differentially coessential in the BRAF-mutant context as compared with BRAF-WT cells (Fig 5B). Consistent with an autonomous signaling role driven by constitutive BRAF activation, in BRAF-mutant cells, downstream MAPK effectors BRAF, MAP2K1 (MEK1), and MAPK1 (ERK2) have markedly reduced fitness correlations with upstream MAPK pathway genes such as EGFR, GRB2, and KRAS (Fig 5B). These findings are made obvious in the context-specific coessentiality networks of major MAPK family members, which depict dense interconnections between all pathway members in BRAF-WT cells but a disconnect between upstream (EGFR and Ras) and downstream (BRAF, MEK, and ERK) genes in BRAF-mutant cells (Fig 5C). Differential network analysis (Ideker & Krogan 2012) directly identifies the relationships selectively lost in BRAF-mutant cells, such as that between EGFR and BRAF, and provides an explanation why pan-cancer

coessentiality analysis fails to identify strong relationships between EGFR/KRAS and downstream MAPK pathway genes (Fig 5C, bottom right panel). Together, these data demonstrate that differential and subset-specific coessentiality network analysis can reveal context-specific genetic interactions undetectable in pan-cancer coessentiality analyses.

## Integration of multiomic data to understand and target contextual gene essentiality

As each cell line in the Project Achilles fitness screening dataset has also undergone extensive multiomic characterization (e.g., transcriptomics, proteomics, and metabolomics), we reasoned that integration of multiomic data could provide insight into the molecular mechanisms underlying genetic dependence on a given factor. In addition, because precision medicine approaches often use -omic data to identify the patients most likely to respond to a given therapy, signatures of gene dependence may be useful in the clinical translation of therapies which target selective genetic dependencies. As a case study, we investigated Heat Shock Factor 1 (HSF1), a multifaceted transcription factor which oversees cytosolic protein homeostasis (Mendillo et al, 2012; Filone et al, 2014; Scherz-Shouval et al, 2014). We built a pan-cancer coessentiality network for HSF1 (Fig 6A), which was enriched for genes involved in protein homeostasis (e.g., HSPA4/Hsp70, DNAJB6/Hsp40, HSPA14/Hsp110, and FKBPL), but also contained genes involved in transcription (e.g., ELP1 and ELP2) and cell cycle regulation (e.g., CENPA and CEP135).

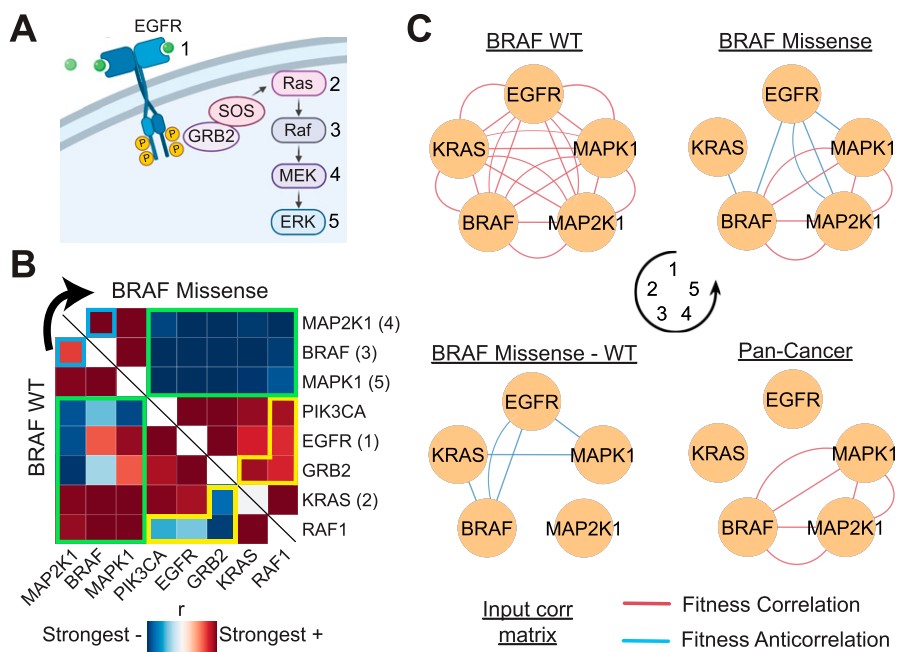

**Figure 5. Context-specific and differential coessentiality network analysis identifies MAPK pathway rewiring by BRAF mutations.**
**(A)** Schematic illustration of several key proteins in the MAPK signaling pathway. Genes in the pathway have multiple paralogs; highlighted paralogs in subsequent graphs for Ras, Raf, MEK, and ERK are KRAS, BRAF, MAP2K1, and MAPK1. **(B)** Ranked Pearson correlations of critical MAPK pathway genes in cell lines without a BRAF mutation (BRAF-WT; n = 643) or with a BRAF missense mutation (n = 96). Ranks are used to make Pearson correlations directly comparable with different sample sizes. Modules featuring differential correlations are highlighted. **(C)** Bottom-up coessentiality networks for select MAPK pathway genes reveals tight interconnections between all pathway members in BRAF-WT cells but a discordant relationship between genes upstream of BRAF in BRAF-mutant lines. Differential network analysis highlights relationships lost in BRAF-mutant cells, such as that between EGFR and BRAF, explaining the obscured signal in pan-cancer coessentiality analysis for these genes.

Next, we built networks for HSF1 within major cellular lineages, finding no substantial enrichment for a given process in any subset (data not shown). Rather, the protein homeostasis genes were consistently correlated with HSF1, suggesting that this canonical role of HSF1 is highly conserved across tissue types and genetic backgrounds (Fig S4).

To better understand the contexts that drive cellular dependence on HSF1, we compared the transcriptome, metabolome, and proteome for cells highly dependent on HSF1 (>75th percentile essentiality) as compared with relatively HSF1-independent cell lines (<25th percentile) across major cancer subsets. Across multiomic signatures, the most striking enrichment involved over-expression of protein synthesis genes in HSF1-dependent cells of several cancer subsets (Fig 6B). This was particularly the case in acute myeloid leukemia (AML), where nearly every gene overexpressed in the HSF1-dependent set of AML lines encoded a protein involved in translation (16 of 23 genes, enrichment $P = 1 \times 10^{-22}$; Fig 6C). Fittingly, protein levels of T308-phosphorylated Akt were elevated in these cell lines, together indicating a high-translation phenotype which does not neatly correlate with known genetic or clinical subtypes of AML. Remarkably, a previous report demonstrated that inhibiting translation initiation with rocaglates inactivates HSF1, suppressing the high-translation malignant state in a manner which most potently impacted AML cells (Santagata et al, 2013). Our data corroborate the observation that protein synthesis demands confer HSF1 dependence in AML, but further suggest that not all AML lines are similarly dependent on this link between HSF1 and translation. Rather, there is a spectrum of this phenotype in AML, which bears clinical consideration given the ongoing exploration of rocaglates/translation initiation inhibitors for the treatment of AML and various other tumors (Cunningham et al, 2018). Indeed, the mRNA signature of translation and HSF1 dependence in AML stratifies AML patients into distinct prognostic groups (Fig 6D).

Together, these data demonstrate the utility of multiomic data integration to better understand the genetic dependencies upon which coessentiality analyses are based and to generate signatures which may be applied to patient samples.

# Discussion

Genes which function in the same biological process often display similar phenotypic variation (e.g., transcript abundance or mutant viability) across biological contexts. As such, the function and interacting partners of individual genes can be predicted by identifying genes with highly correlated phenotypic profiles (Hughes et al, 2000; Dudley et al, 2005). In recent years, high-quality CRISPR-Cas9 screening libraries have provided an unprecedented ability to precisely define the functional consequences of individual gene loss, at genome-scale, in human cells. Fittingly, genome-scale or "top-down" coessentiality network analyses using CRISPR-Cas9 fitness screening datasets have recently emerged as a powerful strategy to identify the functions of and relationships between human genes (Wang et al, 2017; Boyle et al, 2018; Pan et al, 2018; Rauscher et al, 2018; Kim et al, 2019; Wainberg et al, 2019 Preprint; Bayraktar et al, 2020).

Despite their clear value, top-down coessentiality network analyses are most effective assigning function to genes which operate in obligate cooperation with other members of the same process. In turn, many genes—particularly critical regulatory hubs with a complicated network of effectors or roles in multiple biological processes—lack clear module membership or functional enrichment in top-down analyses. On the other hand, we find that bottom-up coessentiality networks centered upon these complicated genes (e.g., RHEB and PPP1R15B) provides clear insight into their functionality. As such, our method for bottom-up network analysis is likely to be particularly useful for investigators

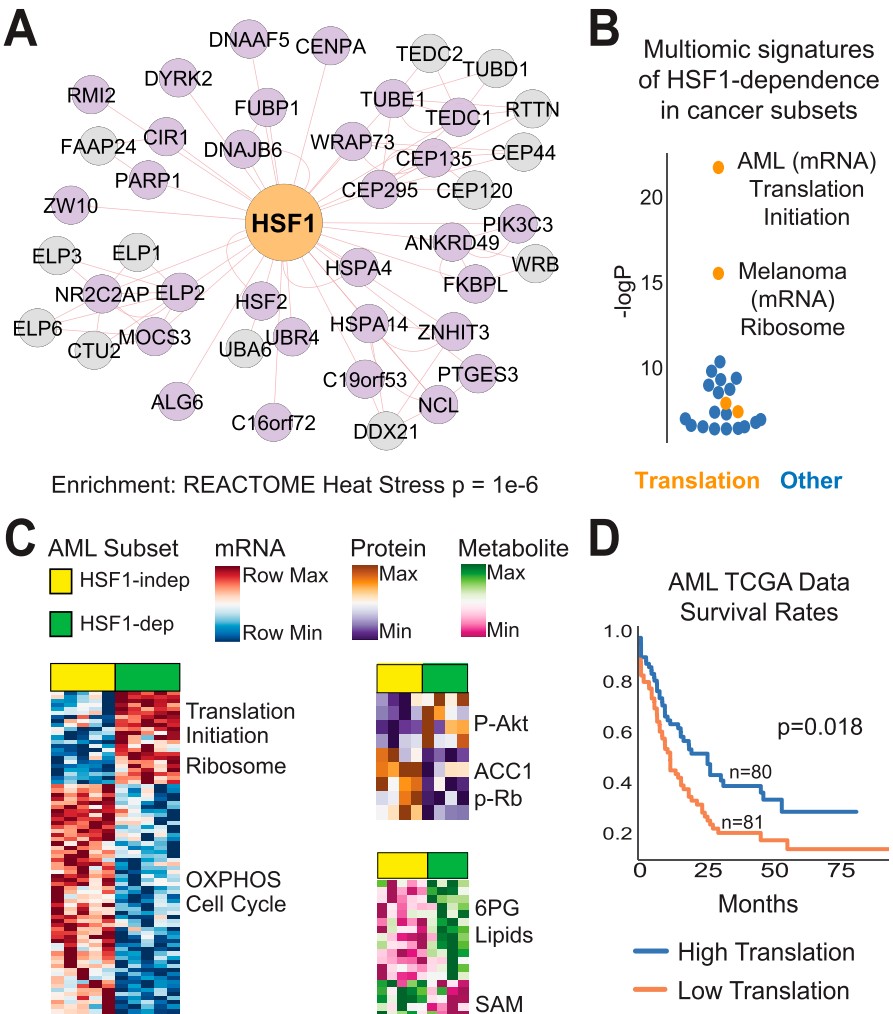

**Figure 6.  Integration of multiomic data reveals increased HSF1 dependence in a biosynthetically active subset of acute myeloid leukemia (AML).** **(A)** The HSF1 coessentiality network, comprising positive connections to rank 30 and 5 potential secondary nodes per gene, is enriched for genes involved in the heat shock cytosolic proteostasis response. *P*-value from hypergeometric test. **(B)** Creation of mRNA, protein, and metabolite signatures of HSF1-dependence (lines with >75th percentile essentiality versus <25th percentile essentiality) across cancer subsets containing at least 10 cell lines. Enrichment *P*-values (hypergeometric overlap test) for the most enriched signatures in each subset are shown. **(C)** Integration of cancer cell line encyclopedia multiomic data to characterize AML cell l`ines stratified by HSF1 dependence (upper versus lower quartile) reveals that HSF1 is most essential in a biosynthetically active subset of AML cell lines. **(D)** The mRNA signature of translation and HSF1-dependence in AML stratifies AML patients into distinct prognostic groups. *P*-value from Cox proportional hazards test.

whose gene(s) of interest do not form robust clusters in published coessentiality resources. However, we emphasize that the bottom-up and top-down approaches are not mutually exclusive. For example, RHEB sits in a highly interconnected module related to mTORC1 in one published coessentiality network (Boyle et al, 2018). If the function of RHEB was not already known, one might create a bottom-up network centered upon RHEB, revealing its nature as a critical intermediate between the TSC1-TSC2 complex and the mTORC1 complex (Fig 3D).

In addition to the source node centric, bottom-up nature of our approach, FIREWORKS offers several novel features and analysis strategies. By developing a preprocessing correction to adjust for locus bias, we reduce the pervasive burden of locus-associated false positive correlations in CRISPR-based coessentiality analysis. Our sliding window correction approach also improved the performance of CRISPR coessentiality to predict genes with known functional and physical interactions. The greatest performance improvements came for genes such as PC and KLF1, whose coessentiality networks before correction contained only genes in the same chromosomal neighborhood, but after correction yielded highly significant enrichment for established partner genes. We emphasize that our approach reduces bias which persists despite

several processing steps designed to mitigate systematic error, as described (Dempster et al, 2019 *Preprint*). In practice, additional approaches to reduce systematic biases in coessentiality analysis include the use of multiple source nodes as the basis for a network investigating a single process. In such a network design (which is easily implemented in our web tool), primary nodes shared between source nodes may be considered the highest confidence associations. In addition, we encourage the use of our secondary node approach to create functional clusters in bottom-up network data. GSEA (Subramanian et al, 2005), encompassing positional gene sets as well as biological pathway gene sets, of these network modules is well equipped to reveal the processes that underlie the shared fitness phenotype within a group of genes, whether positional (uncommon after our locus adjustment approach) or biological.

Another novel feature of our tool lies in the rapid integration of gene-drug interaction data, which we explore in the context of indirect targeting of "undruggable" proteins. Particularly for cancer drivers such as MYC, indirect targeting—that is, modulation of genes which are critical for that driver's effect on cellular signaling but which also can be targeted with small molecules—have yielded

several promising results (Chen et al, 2018). However, the scope of indirect targeting is limited to the small union of genes which are established co-functional partners with a challenging target and which have a druggable structure. By integrating unbiased coessentiality data with a drug–gene interaction database, we found that many pharmacologically challenging targets are putatively co-functional with genes which have known inhibiting or activating drugs. Supporting the validity of these predictions, many indirect targets uncovered in our networks have already demonstrated success in pre-clinical models (e.g., HIF1A-EGLN1, KRAS-RAF1, and MYC-CDK7). In addition, we experimentally validated one novel gene-drug interaction (MYC-PP121, Fig 4E) using a traditional chemical-genetic approach. Given that many genes have 10 or more putative drug interactions in their network, emerging methods to perform multiplexed chemical-genetic profiling may be a useful approach to identify the highest-fidelity indirect targeting strategies (Brockway et al, 2020). Critically, while cancer coessentiality data are most directly applicable to cancer targets, the coessentiality-directed targeting approach may not be limited to oncology. Indeed, proteins of therapeutic interest in other fields, such as the low density lipoprotein receptor (cardiology) or Presenilin-1 (PSEN1, involved in upstream processing of amyloid; neurology) have coessentiality networks enriched for genes involved in their functional pathways, several of which have known drug interactions. Altogether, we propose that integrative coessentiality network analysis is a powerful hypothesis-generating tool which may bolster drug repurposing and precision medicine efforts.

Coessentiality analyses require a sufficiently diverse representation of biological contexts such that most genes have meaningful variations in essentiality across contexts (i.e., greater variation than attributable to experimental error). However, where natural variation is used to achieve contextual breadth, such as in the current practice of using cancer cell line collections, the factors which drive cellular dependence on an individual gene or pathway are difficult to parse. This is in contrast with co-functional approaches where, for example, the same strain of yeast is grown in hundreds of different environmental contexts (Hillenmeyer et al, 2008). We demonstrate that integration of multiomic data from the same cancer cell lines used in coessentiality analysis can provide insight into the contextual factors underlying reliance on a given gene. For example, HSF1 essentiality in AML and melanoma is tightly linked with a protein synthesis phenotype, consistent with a previous report detailing how protein synthesis rates are coupled with HSF1 activation (Santagata et al, 2013; Alasady & Mendillo 2020). Beyond mechanistic considerations, multiomic data may aid in the translation of drug–gene interaction data. Particularly for genes without obvious genomic drivers of essentiality (e.g., mutations or amplifications), attempts to target cells or tumors dependent on a given factor may benefit from multiomic data integration to assist in model selection and patient stratification. For example, if one were to target HSF1 or an HSF1-coessential gene in AML, one would expect tumors with high expression of ribosome and translation genes to be the most likely to respond.

The principal functionality of the FIREWORKS portal leverages pan-cancer coessentiality analysis, which maximizes discovery power by representing the highest number of biological contexts. However, with the understanding that some genetic interactions are context-specific and may not emerge in pan-cancer coessentiality analyses, we provide tools for context-specific network creation. We emphasize that this approach is most suitable for questions of genetic interactions which are specific or differentially regulated in a given context. That is, if an individual studies gene X and works primarily in breast cancer, pan-cancer network analysis will likely provide the best biological insight into gene X, which can then be experimentally tested in breast cancer cells. However, if gene X is normally regulated by a gene universally deleted in breast cancer (or by estrogen, etc.), breast lineage-specific network analysis may reveal alternative regulation mechanisms or other context-specific interactions obscured in pan-cancer analyses. A valid concern with context-specific coessentiality analyses, which draw from a less diverse representation of cell lines, is a reduced power to detect functional interactions. Based on subsampling analyses (Fig S5), we recommend a loose threshold of 12 cell lines per group to minimize this power reduction to the extent possible. Indeed, the first study to demonstrate the power of the CRISPR coessentiality approach used only 14 myeloid leukemia cell lines, but still uncovered several important biological discoveries (Wang et al, 2017). More generally, in all custom coessentiality analyses, the investigator must determine their tolerance for false positives versus false negatives. For example, expanding a network to rank 50 may allow for better resolution of the generalized function of a query gene, whereas restricting a network to rank 5 (or a specified correlation magnitude) will yield only the highest confidence associations, which may be more likely to represent direct functional interactors (e.g., physical interactors).

To facilitate the broad application of bottom-up, integrative coessentiality network analysis, our approach is implemented in an interactive web application (fireworks.mendillolab.org). On the web portal, extensive customization of analysis is possible to facilitate strategies targeted toward a wide spectrum of biological questions. Networks created in the FIREWORKS portal take ~3 s to build and can either be downloaded for further customization in any network analysis/visualization software or saved directly from the web portal as high-resolution, publication-quality images. Critically, as genetic fitness screening data continues to accumulate, the power of bottom-up coessentiality network analysis will only improve. The current iteration of FIREWORKS represents 739 cancer cell lines, and the web portal will be updated with regular feature additions and in parallel with the Broad Institute's DepMap data releases. With several novel analysis strategies and a decreased barrier to access in FIREWORKS, we envision integrative coessentiality analysis becoming a commonplace tool to probe the human genome.

# Materials and Methods

### Public dataset curation and processing

Gene essentiality data derived from CRISPR-Cas9 genome-scale loss-of-function screening of 739 cancer cell lines using a modified Avana library (Doench et al, 2016) as part of Project Achilles was obtained from the Broad Institute's DepMap portal (20q1 release; https://figshare.com/articles/DepMap_20Q1_Public/11791698/3).

CERES scores (Meyers et al, 2017) were used to quantify the fitness effect of individual gene loss, with "essentiality" in this article represented as the CERES score multiplied by −1. For example, a highly essential gene might have a CERES fitness effect of −2 and thus an essentiality score of two. The standard CERES gene effect estimate from the Broad's DepMap portal has undergone several bias adjustments, including removal of confounding principal components, as described (Meyers et al, 2017; Boyle et al, 2018; Dempster et al, 2019 *Preprint*). RNAi gene essentiality data from 712 cancer cell lines, encompassing three independent RNAi screening projects (McFarland et al, 2018), were obtained from (https://figshare.com/articles/DEMETER2_data/6025238/6). Processed RNA-seq, reverse phase protein array, copy number, and metabolomic data were obtained from the DepMap data portal (https://depmap.org/portal/download/). These data are described in detail in Ghandi et al (2019). Cancer cell line drug sensitivity data from the PRISM drug repurposing project are obtainable from https://figshare.com/articles/PRISM_Repurposing_19Q3_Primary_Screen/9393293 (Corsello et al, 2020). Genome positions and chromosomal band annotations for individual genes were obtained from BioMart (Smedley et al, 2009). Duplicate gene family data was downloaded from the Duplicated Gene Database (http://dgd.genouest.org/listRegion/homo_sapiens/all%3A0.x/5/) (Ouedraogo et al, 2012). STRING experimental interactions were obtained from https://string-db.org/cgi/download.pl (Szklarczyk et al, 2019). mRNA co-expression data from COXPRESdb (Obayashi et al, 2019) were downloaded from https://figshare.com/files/10975364. GSEA gene sets were obtained from http://software.broadinstitute.org/gsea/msigdb (Liberzon et al, 2015). CORUM core protein complex member data were obtained from http://mips.helmholtz-muenchen.de/corum/#download (Giurgiu et al, 2019). Drug-gene interaction data was obtained from the Drug-Gene Interaction DataBase at http://www.dgidb.org/data/interactions.tsv (Cotto et al, 2018).

### Quantification of locus bias and Random Forest regression

To determine locus bias for individual genes in CRISPR coessentiality analysis, we assessed each gene's top 100 ranked correlations in the Project Achilles fitness screening dataset and determined the proportion of those correlate genes which were located on the same chromosome. Bias was quantified as observed syntenic coessentiality rate minus expected, where expected was the maximum syntenic coessentiality rate between random chance (determined by number of genes on the chromosome relative to size of the genome) or RNAi coessentiality. To better understand the factors driving locus bias in CRISPR coessentiality data, we trained a machine learning model to predict the bias of individual genes using the following features: gene expression in cancer cell lines (mean and minimum), essentiality score in Achilles CRISPR screens (variance, mean, and maximum), copy number variance of the gene, and its chromosomal band, gene length, and duplicate gene status. This model yielded a decision tree which primarily used three features to predict locus bias with ~17% mean absolute error: local copy number variance, essentiality score variance, and mean gene expression (Fig S1C and D). We chose a Random Forest model because it tolerates feature collinearity and mixed categorical/continuous data and because it allows quantification of the importance of each input feature to the model's predictive success. We found best predictive performance, as determined by mean

absolute error on the test dataset (25% of input data), in a model comprising only the following features: local copy number variance, essentiality score variance of the gene across CRISPR-Cas9 screens, and mean expression of the gene in cancer cell line encyclopedia RNA-seq datasets. Specifically, we used sklearn.ensemble.RandomForestRegressor with n_estimators set to 100. An example tree created with the max_depth parameter set to three was exported for visualization using Graphviz and Pydot.

### Generation of locus-adjusted gene coessentiality matrices

CRISPR Noncorrected refers to the correlation matrix created from the CERES gene effect estimates from the Broad DepMap, which has undergone extensive normalization and denoising as described (Dempster et al, 2019 *Preprint*). Neighbor subtraction, the best-performing locus bias adjustment approach as described in the main text, was performed as follows. The median essentiality score for each gene's 40 nearest neighbors (20 upstream, 20 downstream) was determined and halved. This locus essentiality score was then subtracted from the corresponding gene's initial essentiality estimate before genome-scale, pairwise correlations. Of note, this correction was applied to all genes except for duplicated gene families located within 2.5 MB of each other (n = 3,543), as these genes often have shared functions and were often also correlated in RNAi data. The number of neighbor genes used, as well as other parameters in the neighbor subtraction pipeline, were tested empirically to identify the best-performing version of the neighbor subtraction approach as in Figs 2E–G and S2B and described below. Other locus correction approaches were performed as follows. Biased PCA and Band PCA refer to PCA-based normalization procedures, where PCA was performed on an input matrix and the top principal components were subtracted from the original CERES gene effects matrix before performing correlation analyses. The input matrix for Biased PCA was all genes with substantial locus bias (>75% of coessential genes located on the same chromosome) and for Band PCA was a matrix containing the median essentiality score for chromosomal band across cell lines. The correlation matrices used for benchmarking removed two principal components, but we found no substantial change in performance whether 1, 2, 3, 5, 10, or 15 principal components were used (data not shown). Band Corr Subtract was calculated as follows: to determine the adjusted correlation of Gene X versus Gene Y, the Pearson correlation of Gene X's chromosomal band signature versus Gene Y was subtracted from the correlation of Gene X and Gene Y. Band Removal and Band Penalty refer, respectively, to either removing (setting correlation to zero) or penalizing (dividing correlation coefficient by two) genes in the same chromosomal band region.

### Benchmarking the true discovery rate of different correlation approaches

To assess the ability of different locus bias approaches to predict true positive interactions, we adapted a benchmarking strategy described previously (Pan et al, 2018). Briefly, true positive interactions for a given gene are identified from curated datasets, and the rank at which those genes are identified in each correlation method is obtained. Cumulative distribution functions were then

determined for the identification success rate of true positives as a function of rank threshold. STRING experimental interactions were restricted to high-confidence interactions (score > 0.7). GSEA gene sets included in benchmarking analyses were the following sets: hallmark (H), KEGG pathways (C2), REACTOME (C2), GO Biological Process (C5), and GO Molecular Function (C5). CORUM complexes including only "core" high-confidence complex components.

### Coessentiality network construction and statistical analysis

Rank-based networks were constructed from a single or set of input genes, using a soft rank threshold for each analysis, that is, correlations below the specified rank were not included. Edges are not weighted by correlation strength or rank. Networks were visualized in Cytoscape v3.7.2 (https://cytoscape.org/) (Shannon et al, 2003). Networks used a standard force-directed layout with manual adjustments made where needed to improve legibility. As described in the main text, control/simulated genes were created by randomly sampling sets of 739 gene effect estimates from the Project Achilles dataset. The rationale for this approach is that, by choosing a rank-based network creation approach, every gene (independent of the biological validity of its signal-to-noise ratio in the DepMap dataset) will have the same number of fitness correlations at a given rank threshold. Thus, by creating genes strictly from noise, we could determine the magnitude of correlation coefficient which could be explained by chance at a given rank threshold and false discovery rate cutoff. Similarly, because a "noise" gene may (by chance) have a similar fitness profile to a highly interconnected set of genes, such as components of a large molecular assembly (e.g., the mitochondrial ribosome), some networks would also be expected to have high numbers of internal edges (connections between non-isolated genes in the network) strictly as a function of chance. Thus, we determined the number of internal edges (here, defined as edges between nodes which have more than one total connection in the network) present in each "noise" network as a backdrop to identify networks with truly enriched functional modules. Louvain modularity (Blondel et al, 2008) for each bottom-up network was determined using the Python modules Networkx and Python-Louvain.

### Drug-gene interaction data integration

Because not all interactions in the Drug-Gene Interaction DataBase (DGIDB; Cotto et al, 2018) have an annotated mechanism, these interactions were excluded in all analyses where direct MOA is reported. MOAs were classified as follows, with bracketed categories being the term annotated in DGIDB: inhibitor (gating inhibitor, inhibitor, blocker, antisense oligonucleotide, antagonist/inhibitor, allosteric modulator/antagonist, vaccine, partial agonist, antagonist, antibody, channel blocker/gating inhibitor, inverse agonist, negative modulator, suppressor, channel blocker, antisense, inhibitory allosteric modulator, and activator/channel blocker), activator (stimulator, activator, agonist, cofactor, inducer, agonist/allosteric modulator, activator/antagonist, and positive allosteric modulator), or other (binder, agonist/antagonist, allosteric modulator, and modulator). We note that interactions classified as other often comprised complex interactions, for example, selective modulation of the estrogen receptor by tamoxifen.

"Both" indicates that a given gene/protein has an activating and an inhibitory drug interaction. In the FIREWORKS web portal, only interactions with an annotated mechanism are reported.

### Cellular viability assay

Cell viability was estimated using the colorimetric, WST-8 tetrazolium salt-based, Cell Counting Kit-8 (CCK-8) from Dojindo Molecular Technologies (SKU: CK04). For HO15.19 rat fibroblast cells (MYC KO), 2,000 cells/well were seeded in 96-well plates. For TGR-1 (parental MYC WT), 1,000 cells/well were seeded in 96-well plates. 3 d after treatment with PP121 (SelleckChem S2622), CCK-8 solution was added, and plates incubated for 3 h. Absorption was measured at 450 nm using plate reader (Perkin Elmer Victor 3V).

### Cell line multiomic data integration and descriptive comparisons

For descriptive comparisons of different cell lines stratified by dependency signatures, the cell lines with 75th percentile or higher dependency on that gene were compared with cell lines having 25th percentile or lower dependency. Subsets of cancer cell lines with at least 15 cell lines were considered for subset multiomic analyses to mitigate false positive discoveries from underpowered analyses. Enrichment of signatures, as computed for HSF1-associated multiomic signatures, was determined using a hypergeometric test of gene overlap between GSEA gene sets and the signature genes/proteins. Because our primary end point in multiomic signature analysis is in GSEA or further development of a patient gene signature, differential abundance was assessed by a simple two-tailed $t$ test with a lenient significance threshold of 0.005 (transcriptomic) or 0.05 (proteomic and metabolomic) data. However, we note that $P$-value thresholds are customizable in the FIREWORKS portal and should be considered in the context of the number of cell lines included in a subset analysis.

### Analysis of AML patient survival

Log-normalized RNA-sequencing data were obtained for AML patients from The Cancer Genome Atlas Pan-Cancer Atlas as accessible at cBioPortal (Cerami et al, 2012). Of 200 patients, 161 had mRNA abundance estimates available for genes in the high-translation signature which were overexpressed in HSF1-dependent AML cell lines (EIF3L, RPL3, EEF1D, RPL34, FNBP4, RPS13, RSPH4A, RPS12, BRD8, CCNI, RPL27, RPL32, RPS3A, RPL10, RPL7A, EEF1A1, RPS14, USP38, RPS23, ZNF33B, HSD17B11, RPS29, and EEF1G). The median log-RSEM of these genes was taken for each case as the value of the mRNA translation signature, and patients were stratified at the median expression value for the signature for survival analysis. Kaplan–Meyer plotting and statistical analysis (Cox proportional hazards test) were performed in the Python package Lifelines v0.24.9.

### Gene set enrichment analysis

GSEA was performed using the Molecular Signature Database as accessible at http://software.broadinstitute.org/gsea/msigdb/annotate.jsp (Subramanian et al, 2005; Liberzon et al, 2015). The gene sets queried

were as follows: hallmark (H), positional (C1), KEGG pathways (C2), REACTOME (C2), GO Biological Process (C5), and GO Molecular Function (C5). The ranked gene set enrichment plots in Fig 2H were made in Python using a modified version of the seaborn.rugplot function in Python, with enrichment *P*-values calculated using a two-sample Kolmogorov-Smirnov test implemented in scipy.stats.ks_2samp.

## Data and code availability

All codes used in this manuscript are available at https://github.com/mendillolab. Our interactive web application to construct and integrate bias-adjusted coessentiality networks for a given set of input genes is available at https://fireworks.mendillolab.org.

# Supplementary Information

# Acknowledgements

We thank Drs Daniel R Foltz, Eva Gottwein, and Mark Manzano for insightful discussions. We thank Bettina H Cheung for comments on the manuscript. DR Amici was supported by 5T32GM008152-33. ML Mendillo is supported by the National Cancer Institute of the National Institutes of Health (R00CA175293) and the Susan G Komen Foundation (CCR17488145). ML Mendillo was also supported by Kimmel Scholar (SKF-16-135) and Lynn Sage Scholar awards.

## Author Contributions

DR Amici: conceptualization, resources, data curation, software, formal analysis, validation, investigation, visualization, methodology, and writing—original draft, review, and editing.
JM Jackson: resources, data curation, software, formal analysis, validation, investigation, visualization, methodology, and writing—review and editing.
MI Truica: data curation, formal analysis, investigation, methodology, and writing—review and editing.
RS Smith: data curation, validation, and writing—review and editing.
SA Abdulkadir: resources, supervision, methodology, and writing—review and editing.
ML Mendillo: conceptualization, supervision, funding acquisition, methodology, project administration, and writing—review and editing.

## Conflict of Interest Statement

The authors declare that they have no conflict of interest.

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
