## [Reviewer comments · Life Science Alliance]

Life Science Alliance

FIREWORKS: a bottom-up approach to integrative coessentiality network analysis

David Amici, Jasen Jackson, Mihai Truica, Roger Smith, Sarki Abdulkadir, and Marc Mendillo
DOI: <https://doi.org/10.26508/lsa.202000882>

Corresponding author(s): Marc Mendillo, Northwestern University Feinberg School of Medicine

Review Timeline:

Submission Date:	2020-08-17
Editorial Decision:	2020-09-25
Revision Received:	2020-11-12
Editorial Decision:	2020-11-30
Revision Received:	2020-12-01
Accepted:	2020-12-02

Scientific Editor: Shachi Bhatt

Transaction Report:

September 25, 2020

Re: Life Science Alliance manuscript #LSA-2020-00882

Marc L Mendillo
Northwestern University Feinberg School of Medicine

Dear Dr. Mendillo,

Thank you for submitting your manuscript entitled "FIREWORKS: a bottom-up approach to integrative coessentiality network analysis" to Life Science Alliance (LSA). The manuscript has been reviewed by the editors and outside referees (reviewer comments below). As you will see, the reviewers were very enthusiastic about the study and its potential impact, and have raised minor concerns that should be addressed prior to further consideration of the manuscript at LSA. Therefore, although we are unable to publish the current version of the manuscript, we would kindly encourage you to submit a revised version that addresses the referees' concerns, particularly the concern pointed out by Reviewer 2 that some of the analyses discussed in the manuscript are not appear implemented in the tool.

We would be happy to discuss the individual revision points further with you, should this be helpful. A revised manuscript may be re-reviewed, most likely by some or all of the original referees. When submitting the revision, please include a letter addressing the reviewers' comments point-by-point and a copy of the text with alterations highlighted (boldfaced or underlined). The typical time frame for revisions is three months. In an effort to expedite the review process, papers are generally considered through only one revision cycle.

While you are revising your manuscript, please also attend to the below editorial points to help expedite the publication of your manuscript.

Thank you for considering LSA as an appropriate venue for your research. We look forward to receiving your revised manuscript.

Sincerely,

Shachi Bhatt, Ph.D.
Executive Editor
Life Science Alliance

B. MANUSCRIPT ORGANIZATION AND FORMATTING:

Reviewer #1 (Comments to the Authors (Required)):

Amici et al. presents FIREWORKS, a tool that provides a user-friendly interface to explore co-essentiality networks. It is particularly interesting their statistical normalisation step in the creation of the co-essentiality networks that allows to reduce the false positive rates related with genes that are genomically close, i.e. syntenic co-essentiality.

This is an important, but often neglected aspect, especially when using CRISPR-Cas9 loss-of-function screens. The manuscript is very well written and presented, with clear examples that benefit from the correction, computational analyses are well performed and explained, and the webtool is intuitive and robust.

I only have a couple of comments:

1. Several methods to correct copy number deleterious bias of CRISPR-Cas9 screens are currently available. Could the authors clarify if they have used CERES or CRISPRcleanR corrected CRISPR-Cas9 screens? Do they see such strong syntenic co-essentiality bias even after correction? This is particularly relevant for the general analysis of CRISPR-Cas9 screens, since it would be important to understand if copy number corrected fold changes are still affected by these effects.

2. As the authors state, co-essentiality networks can be used to explore drug mechanisms-of-action. The use of both RNAi and CRISPR-Cas9 screens is particularly interesting. Have the authors performed differential network analysis between RNAi and CRISPR co-essentiality networks around the same drug target? Could this be used to inform on drug-targets where inhibition is intrinsically different than knockout (e.g. PARP1/2)?

Reviewer #2 (Comments to the Authors (Required)):

Summary:

Amici and colleagues present Fireworks, an online tool for building and exploring co-essentiality networks derived from CRISPR screens in panels of cancer cell lines. As the authors note, co-essentiality networks derived from CRISPR screens have already been shown to provide insight into gene function, but there are few tools to exploit co-essentiality networks that can be used by those without computational skills. Consequently the tool should be very useful to the community. The authors' 'bottom-up' approach is a nice way to avoid the resulting co-essentiality networks being dominated by largely stable protein complexes. The authors also develop new approaches to avoid co-essentiality networks being biased by chromosome location effects. Overall the tool and the authors' approach appear very useful but the manuscript could be a little clearer and more focused.

Although the stated purpose of the manuscript is to 'demonstrate the broad utility of FIREWORKS through case vignettes' some of the analyses discussed do not appear implemented in the tool. For example, the use of 'differential co-essentiality' (Fig. 5 and associated discussion) seems both novel and useful, but does not seem to be implemented in the online tool. Similarly Figure 4 and the associated discussion highlight the integration of drug-gene interaction data with co-essentiality networks to identify targets for 'undruggable' proteins. However this does not seem readily supported by the online tool - i.e. it does not appear possible to highlight all the druggable targets within a given co-essential network. This could be addressed by focusing the manuscript on analyses that can be performed using the online tool or by implementing the described functionality in the online tool.

Specific comments:

It's not clear to me what the relationship is between this manuscript and a preprint from the same authors (<https://doi.org/10.1101/847996>) which also presents the fireworks tool.

Boyle et al (Mol Syst Bio 2018) previously developed methods to correct systematic biases in co-essential networks derived from CRISPR screens. These should at least be discussed in comparison with the approaches of Amici et al.

Page 4 - typo 'siRNA' (should read shRNA)

page 7/8 - why 40 genes? why 1/2 the median? were these numbers determined empirically or how were they identified?

Figure 2C - what's on the y-axis?

Figure 4 D - what is the r value? Should be shown alongside p-value

page 15 - does matrix subtraction just mean the difference between the two correlation matrices?

page 22 - is there any empirical basis for a threshold of 15 cell lines? it would be relatively straightforward to subsample cell lines and see at what point reasonable signal is obtained.

Fig 6 - this figure and the associated discussion of multiomic data appear largely orthogonal to the rest of the paper. The text primarily relates to biomarkers of HSF1 sensitivity rather than co-essentiality.

Dear Dr. Bhatt,

We thank you and the reviewers for carefully assessing our manuscript, entitled "FIREWORKS: a bottom-up approach to integrative coessentiality network analysis" by Amici et al. (LSA-2020-00882).

We were delighted by the very positive comments of the reviewers on our initial submission. We are excited to return our manuscript that addresses all reviewer concerns. In particular, we have made substantial improvements to the web tool. Networks now load approximately 10-fold faster, "druggable" genes can be toggled with ease, and custom correlation matrices (e.g. differential matrices) can be used in our network building functions. Additionally, in the manuscript, we added an experimental validation for the indirect interaction between MYC and WNK1 inhibition (Figure 4E). All of these and other minor concerns are detailed below in our point-by-point response to the reviewers' original comments (in blue).

Reviewer #1 (Comments to the Authors (Required)):

Amici et al. presents FIREWORKS, a tool that provides a user-friendly interface to explore co-essentiality networks. It is particularly interesting their statistical normalisation step in the creation of the co-essentiality networks that allows to reduce the false positive rates related with genes that are genomically close, i.e. syntenic co-essentiality.

This is an important, but often neglected aspect, especially when using CRISPR-Cas9 loss-of-function screens. The manuscript is very well written and presented, with clear examples that benefit from the correction, computational analyses are well performed and explained, and the webtool is intuitive and robust.

I only have a couple of comments:

1. Several methods to correct copy number deleterious bias of CRISPR-Cas9 screens are currently available. Could the authors clarify if they have used CERES or CRISPRcleanR corrected CRISPR-Cas9 screens? Do they see such strong syntenic co-essentiality bias even after correction? This is particularly relevant for the general analysis of CRISPR-Cas9 screens, since it would be important to understand if copy number corrected fold changes are still affected by these effects.

The reviewer makes the important point that algorithms (e.g. CERES, CRISPRcleanR) have been designed to mitigate bias introduced by Cas9-mediated cleavage of copy number-variable regions. The input or "pre-locus-correction" CRISPR gene essentiality dataset we use in this manuscript has already undergone CERES correction (copy-number-based correction of the magnitude of estimated gene effect) as well as an additional bias removal step, Removal of Confounding Principal Components (RCPC; an optimized variation of the PCA bias removal technique described in Boyle, 2018 and mentioned by reviewer 2) [1]. Remarkably, the syntenic coessentiality bias described in our article persists despite these bias-reduction approaches, indicating the need for additional correction (as we describe) for modified coessentiality analysis. We have modified the text to clarify this point, including the results section; underlined is changed:

"We next identified the proportion of each gene's top 100 ranked fitness correlations which are located on the same chromosome using CRISPR-Cas9 gene essentiality estimates derived from the Broad Institute's Dependency Map screening project of 739 cancer cell lines (Meyers et

al. 2017; Tsherniak et al. 2017) (Figure 2A). Importantly, these CRISPR-based essentiality estimates have already undergone several bias reduction steps, including application of the CERES algorithm (which adjusts the gene effect estimate based upon local copy number) and PCA-based denoising similar to that described by Boyle et. al (Meyers et al. 2017; Boyle et al. 2018; Dempster et al. 2019). Despite these preprocessing steps, we found...

2. As the authors state, co-essentiality networks can be used to explore drug mechanisms-of-action. The use of both RNAi and CRISPR-Cas9 screens is particularly interesting. Have the authors performed differential network analysis between RNAi and CRISPR co-essentiality networks around the same drug target? Could this be used to inform on drug-targets where inhibition is intrinsically different than knockout (e.g. PARP1/2)?

This is an intriguing application of the differential coessentiality concept. We had not previously performed differential analyses between RNAi and CRISPR datasets, but we performed a preliminary analysis of PARP1 and PARP2 as suggested by the reviewer. Suggesting there may be intriguing biology uncovered by this approach, we found that PARP10 was the 5th most differentially coessential gene with PARP2 in the RNAi setting (RNAi correlation, 0.27; CRISPR corrected correlation, -0.15). Underscoring the RNAi data is always the problem of off-target effects driven by dominant seed effects [2, 3]. Thus, it is possible that some hairpin(s) targeting PARP2 also target PARP10. However, it is also possible that PARP10 (and other differentially coessential proteins) represent a true synthetic vulnerability in cells under acute loss of PARP2, but not PARP2 knockout. Future studies in this area may reveal interesting biology – for this reason, we have now made it possible to perform this type of analysis in our updated web tool (upload custom coessentiality matrix).

Reviewer #2 (Comments to the Authors (Required)):

Summary:

Amici and colleagues present Fireworks, an online tool for building and exploring co-essentiality networks derived from CRISPR screens in panels of cancer cell lines. As the authors note, co-essentiality networks derived from CRISPR screens have already been shown to provide insight into gene function, but there are few tools to exploit co-essentiality networks that can be used by those without computational skills. Consequently the tool should be very useful to the community. The authors' 'bottom-up' approach is a nice way to avoid the resulting co-essentiality networks being dominated by largely stable protein complexes. The authors also develop new approaches to avoid co-essentiality networks being biased by chromosome location effects. Overall the tool and the authors' approach appear very useful but the manuscript could be a little clearer and more focused.

Although the stated purpose of the manuscript is to 'demonstrate the broad utility of FIREWORKS through case vignettes' some of the analyses discussed do not appear implemented in the tool. For example, the use of 'differential co-essentiality' (Fig. 5 and associated discussion) seems both novel and useful, but does not seem to be implemented in the online tool. Similarly Figure 4 and the associated discussion highlight the integration of drug-gene interaction data with co-essentiality networks to identify targets for 'undruggable' proteins. However this does not seem readily supported by the online tool - i.e. it does not appear possible to highlight all the druggable targets within a given co-essential network. This could be addressed by focusing the manuscript on analyses that can be performed using the online tool or by implementing the described functionality in the online tool.

The reviewer brings up the point that certain analysis strategies presented in the manuscript were not accessible in the web tool. As suggested by the reviewer, we have updated the web tool to encompass the breadth of analyses discussed in the manuscript. Specifically, we have added functionality for differential (and other custom) coessentiality network analyses and easier toggling of “druggable” genes in a given network.

Specific comments:

It's not clear to me what the relationship is between this manuscript and a preprint from the same authors (<https://doi.org/10.1101/847996>) which also presents the fireworks tool.

This early version of our work introduced the first version of our bottom-up coessentiality approach as applied to stress response biology (an interest of our lab). Our approach and the accompanying tool has undergone significant expansion and optimization since posting of this preprint, which led us to seek publication of this article which focuses on the method/tool itself. We emphasize that no overlapping data is or will be under consideration elsewhere.

Boyle et al (Mol Syst Bio 2018) previously developed methods to correct systematic biases in co-essential networks derived from CRISPR screens. These should at least be discussed in comparison with the approaches of Amici et al.

The reviewer brings up a coessentiality bias-reduction approach based upon PCA. We did not discuss this approach in our original draft because our baseline, pre-locus-correction CRISPR dataset has already undergone a PCA-based denoising procedure (Removal of Confounding Principal Components; RCPC) based on that described by Boyle et. al. Thus, our approach is not an alternative, but rather complementary, to the RCPC concept which has already become standard in the Broad Institute's DepMap pipeline [1]. As mentioned above, in the revised manuscript, we clarify that PCA-based denoising as first described in Boyle et. al has been performed on the Broad DepMap's CRISPR-Cas9 gene effect data.

Page 4 - typo 'siRNA' (should read shRNA)

We have corrected this error.

page 7/8 - why 40 genes? why 1/2 the median? were these numbers determined empirically or how were they identified?

In the methods section, we noted that “the number of neighbor genes used, as well as other parameters in the neighbor subtraction pipeline, were tested empirically to identify the best-performing version of the neighbor subtraction approach as in Figures 2E-G and S2B and described below.” However, this was not mentioned in the results and could be further emphasized, thus we have edited the results as shown below:

“...for each pre-correction gene not located within a duplicate gene cluster, half the median essentiality score of 40 neighbor genes is subtracted from the pre-correction gene's initial essentiality estimate (Figure S2C). We note that these adjustment parameters (e.g. number of neighbors) were determined through unbiased benchmarking (Methods). The sliding window correction approach serves to effectively smooth out fitness effects...”

Figure 2C - what's on the y-axis?

Feature importance here was defined as the mean decrease in impurity, also called the gini importance. The figure legend has been updated to clarify.

“(C) Gini importance, a measure of the power of a feature to reduce model uncertainty, of gene-level features in a Random Forest regression model trained to predict locus bias”

Figure 4 D - what is the r value? Should be shown alongside p-value

We have updated the figure to include r values.

page 15 - does matrix subtraction just mean the difference between the two correlation matrices?

Yes. We have reworded this sentence to improve clarity.

“To identify coessential relationships specific to or enriched in the BRAF-mutant context, we generated a differential correlation matrix by subtracting the BRAF-WT matrix from the BRAF-mutant matrix.”

page 22 - is there any empirical basis for a threshold of 15 cell lines? it would be relatively straightforward to subsample cell lines and see at what point reasonable signal is obtained.

This general recommendation for lineage-specific (or other subset analysis) was based on experience and not empiric testing. To make a more robust suggestion, we chose a lineage (bone; 29 cell lines) to subsample. We chose one lineage instead of sampling the whole collection of cell lines because the genomic and phenotypic heterogeneity within a lineage is expected to be less than the heterogeneity of all lines in the collection – and thus, intra-lineage analyses may be expected to require more cell lines to achieve sufficient power. We performed 50 samplings of the bone cancer essentiality dataset at each threshold from 3 to 28 cell lines and compared the resulting correlation matrix with the original bone correlation matrix.

We assessed two outcomes which focused on the ability to detect strong positive correlations (rank 1-50): success rate, a binary variable defined as the proportion of genes with well-resolved networks (rank position in the same decile) in the subset dataset, and mean error, a continuous variable defined as the mean difference in rank position for genes in full-sample network as compared with the subsample dataset. We found an inflection point in both outcomes around 12-15 cell lines. We have added a supplementary figure (reproduced below for convenience) which shows this analysis and we have edited the recommended number of cell lines in the text to 12 (versus 15).

Fig 6 - this figure and the associated discussion of multiomic data appear largely orthogonal to the rest of the paper. The text primarily relates to biomarkers of HSF1 sensitivity rather than co-essentiality.

We agree that this feature is the most divergent from our core network approach. However, as mentioned in the discussion, where natural variation is used to find coessential patterns, multiomic data may help to determine the factors which may drive dependence on a given gene – and thus better understand and apply coessentiality findings derived from our tool. For example, our result suggests that HSF1 is co-functional with protein folding genes independent of cellular context, and that targeting any members of its network may be efficacious where

HSF1 contributes to disease. However, the network itself can not inform which tumors (in the case of cancer) are most likely to respond to such therapy. In this sense, multiomic integration reveals that i) AMLs with high expression of protein synthesis genes may be more likely to respond to HSF1 network therapies and, perhaps equally importantly, ii) protein synthesis gene expression may not be a suitable approach to estimate HSF1 network dependence in several other types of cancer.

REFERENCES

1. Dempster, J.M., et al., *Extracting Biological Insights from the Project Achilles Genome-Scale CRISPR Screens in Cancer Cell Lines*. bioRxiv, 2019: p. 720243.
2. Singh, S., et al., *Morphological Profiles of RNAi-Induced Gene Knockdown Are Highly Reproducible but Dominated by Seed Effects*. PLoS One, 2015. **10**(7): p. e0131370.
3. Smith, I., et al., *Evaluation of RNAi and CRISPR technologies by large-scale gene expression profiling in the Connectivity Map*. PLoS Biol, 2017. **15**(11): p. e2003213.

November 30, 2020

RE: Life Science Alliance Manuscript #LSA-2020-00882R

Dr. Marc L Mendillo
Northwestern University Feinberg School of Medicine
303 E. Superior St., Room 7-303
Chicago, Illinois 60610

Dear Dr. Mendillo,

Thank you for submitting your revised manuscript entitled "FIREWORKS: a bottom-up approach to integrative coessentiality network analysis". We would be happy to publish your paper in Life Science Alliance pending final revisions necessary to meet our formatting guidelines.

Along with the points listed below, please also attend to the following,

- please add ORCID ID for corresponding author-you should have received instructions on how to do so
- please add a separate section with your Figure Legends-both main and supplementary-under the reference section

A. FINAL FILES:

B. MANUSCRIPT ORGANIZATION AND FORMATTING:

Sincerely,

Shachi Bhatt, Ph.D.
Executive Editor
Life Science Alliance
<https://www.lsjournal.org/>
Tweet @SciBhatt @LSAJournal

Reviewer #1 (Comments to the Authors (Required)):

The authors have addressed all my comments satisfactorily.

Reviewer #2 (Comments to the Authors (Required)):

The authors have addressed all of my concerns. The approach developed and the shiny app should be very useful for the community.

December 2, 2020

RE: Life Science Alliance Manuscript #LSA-2020-00882RR

Dr. Marc L Mendillo
Northwestern University Feinberg School of Medicine
303 E. Superior St., Room 7-303
Chicago, Illinois 60610

Dear Dr. Mendillo,

Thank you for submitting your Research Article entitled "FIREWORKS: a bottom-up approach to integrative coessentiality network analysis". It is a pleasure to let you know that your manuscript is now accepted for publication in Life Science Alliance. Congratulations on this interesting work.

DISTRIBUTION OF MATERIALS:

Again, congratulations on a very nice paper. I hope you found the review process to be constructive and are pleased with how the manuscript was handled editorially. We look forward to future exciting submissions from your lab.

Sincerely,

Shachi Bhatt, Ph.D.

Executive Editor

Life Science Alliance

<https://www.lsjournal.org/>
